**Research**

# A national survey of Russian physicians' knowledge of diagnosis and management of food-induced anaphylaxis

Daniel Munblit,[1,2,3] Marina Treneva,[3,4] Ilya Korsunskiy,[2] Alan Asmanov,[4] Alexander Pampura,[4] John O Warner[1,3,5]

► Prepublication history and additional material are available. To view these files, please visit the journal online (http://dx.doi.org/10.1136/bmjopen-2017-015901).

[1]Department of Paediatrics, Imperial College London, London, UK
[2]Faculty of Pediatrics, I.M. Sechenov First Moscow State Medical University, Russia
[3]International Inflammation (in-FLAME) Network of the World Universities Network, Perth, WA, Australia
[4]Allergy Department, Veltischev Clinical Pediatric Research Institute of Pirogov Russian National Research Medical University, Russia
[5]National Institute of Health Research Collaboration for Leadership in Applied Health Research and Care for NW London, London, UK

**Correspondence to**
Dr Daniel Munblit; daniel.munblit08@imperial.ac.uk

## ABSTRACT

**Objectives** Food allergy is an increasing burden worldwide and is a common problem within paediatric populations, affecting 5%–8% of children. Anaphylaxis caused by food proteins is a potentially life-threatening condition and all healthcare practitioners should be aware of its recognition and management. Russia is the largest country in Europe but it is still unknown whether physicians are prepared to diagnose and manage food-induced anaphylaxis effectively. We aimed to examine physicians' knowledge of diagnosis and management of food-induced anaphylaxis.

**Setting, population and outcomes** A survey was designed and published online at VrachiRF.ru website (for registered Russian-speaking practicing physicians). We obtained information on respondents' clinical settings, experience and specialty. Survey questions were based on a characteristic clinical scenario of anaphylaxis due to food ingestion. Outcome measures consisted of correct answers to the anaphylaxis diagnosis and management questions.

**Results** From a total of 707 of physicians accessed in the survey, 315 (45%) responded to the clinical scenario. 16 respondents reported training in allergy-immunology and have been excluded from the analysis, leaving the final sample size of 299. Respondents were paediatricians (68%) and other specialties adult physicians (32%). Overall, 100 (33%) of respondents diagnosed anaphylaxis, but only 29% of those making the correct diagnosis administered adrenalin (1:1000) intramuscular. Respondents working in secondary/tertiary clinics diagnosed anaphylaxis significantly more often (p=0.04) when compared with primary care/private practice physicians. This difference was also apparent as the most important influence on responses in the multivariate analysis.

**Conclusions** In this national sample of Russian physicians, we found poor knowledge in both anaphylaxis diagnosis and management. Our data show that the chance of being properly diagnosed with anaphylaxis is 33% and being appropriately treated with adrenalin is 10%. These findings highlight lack of anaphylaxis knowledge among Russian physicians, both paediatricians and other specialists and illustrates the urgent need for allergy/anaphylaxis training.

### Strengths and limitations of this study

► Strengths of this survey include the large number of respondents from different medical backgrounds. This is the first survey of anaphylaxis knowledge in Russian physicians and provides an insight into current ability of Russian doctors to diagnose and manage anaphylaxis. As all surveyed physicians answered the same set of questions, it allows for comparisons of their knowledge between the groups.

► Limitations of the study are the use of non-standardised and validated survey instruments. The distribution via VrachiRF.ru website where respondents may be more educated than average physicians, and the 45% response rate probably present a best rather than worst-case scenario.

## INTRODUCTION

Allergic conditions are a major problem worldwide and their prevalence have increased considerably in the recent decades with a quarter of school-age children in Europe being allergic. Among allergic diseases, food allergy has become a significant burden with 6%–8% of children under the age of 3 years having allergy to at least one food.[1] Food allergy has detrimental effects on the quality of life[2] and health/economics with prescribing of adrenalin auto-injectors (AAIs) having risen over 350-fold in the UK from 2000 to 2012.[3]

Anaphylaxis is a 'severe, life-threatening generalised systemic hypersensitivity reaction'.[4] Details on anaphylaxis prevalence, the main triggers and mortality rates are available in some countries, but missing in many geographical locations. A recent meta-analysis undertaken by the European Academy of Allergy and Clinical Immunology (EAACI) provided an estimated pooled prevalence of anaphylaxis of 0.3%.[5] No data on anaphylaxis

prevalence and/or mortality rates are available from the Russian Federation. The reasons for the lack of research in this field in Russia remain unclear, but may be partially explained by lack of a unified digital registry of anaphylaxis events and unavailability of AAIs, as none of the existing brands is registered in Russian Federation. The registration is a prerequisite for a drug to be approved for clinical use. The process of approval for medical use of a drug includes drug quality, efficacy, safety and, in case with autoinjectors, delivery, assessment by an authorised state body.

Food allergy is a major health concern and is increasing in prevalence reaching an overall pooled point prevalence of 0.9% when confirmed by oral food challenge.[6] Data from the USA shows that food-induced anaphylaxis is a cause of 2000 hospitalisations and 200 deaths a year.[7] Food is the leading cause of anaphylactic reactions, especially in paediatric populations[8 9] and when associated with coexisting asthma has an increased risk of death.[10] Coexistence of food allergy and asthma is not an unusual event as 4%–8% of asthmatic children with asthma have food allergy, and more than a third of children with food allergy have asthma.[11] The most recent UK-wide investigation on 195 asthma deaths found that many people died of asthma with coexistent allergy between 2012 and 2013.[12]

EAACI has underlined the importance of anaphylaxis recognition as a clinical emergency, and highlighted the need for healthcare professionals to be aware of its diagnosis and management.[4] The EAACI taskforce on anaphylaxis drafted guidelines covering all aspects of recognition, risk factor assessment and the management of patients. Despite these efforts, levels of anaphylaxis recognition and management knowledge is still far from perfect.[7 13]

The Russian Association of Allergy and Clinical Immunology (RAACI) local guidelines provide physicians with the basic principles of anaphylaxis diagnosis and management,[14 15] but the degree of anaphylaxis understanding among Russian physicians is unknown. Our survey aimed to provide information of Russian physicians' knowledge of anaphylaxis diagnosis and management.

## METHODS
### Study design and population
A cross-sectional study was conducted in January 2016 using an online survey published on the VrachiRF ( VrachiRF.ru) website. VrachiRF is a popular Russian professional website for registered physicians (with about half a million subscribers), which provides clinically relevant information and latest news in the medical field. The website sends daily emails to all the members of VrachiRF community with the latest medical news and updates and a special email on Sunday, providing members with the top news of the week. VrachiRF placed a survey on the website for 4 weeks (between February and March 2016), responses were collected

and anonymised data presented in an Excel spreadsheet. The study population included paediatricians and adult physicians of various specialities. The survey was conducted following ethical approval by Moscow Clinical Hospital №9, named after G.N. Speransky, Ethics Committee.

### Survey questionnaire
We developed a case-based survey in Russian to evaluate physicians' knowledge of anaphylaxis diagnosis and management. The survey consisted of a clinical scenario; open-ended questions on diagnosis; multiple-choice questions on treatment, required clinical tests and recommendations. The clinical scenario described an anaphylactic event with multiple system involvement.

During a busy weekend A&E shift, you are seeing a boy aged 6 years. His main complaint at admission is syncope. Ten minues after eating a Chinese meal at a friend's house, he developed an urticarial rash on the neck. Five minutes after the rash appearance, he complained of abdominal cramping and vomited twice. After 15 min, he felt dizzy, developed a wheeze and fainted. He was delivered to the hospital 35 min after occurrence of the first symptoms. It is not known if he has food allergy but he had two hospitalisations for asthma in the last year.

On examination, his heart rate was 90, blood pressure 90/45 mm Hg, respiratory rate 22 and oxygen saturation 94%. His weight was 22 kg. He had diffuse wheezing and was in moderate respiratory distress. He was not carrying his salbutamol inhaler and had not received any medications yet.

As no validated survey instrument exists, the development of this questionnaire was based on allergy expert opinion and previously published data. The survey was developed by the research team, consisting of paediatricians and allergy specialists. It was pilot tested with external adult physicians, paediatricians and allergy specialists for clarity, prior to its use.

The first question asked responders to provide the most likely diagnosis for the clinical scenario in an open-answer form. The next page of online survey provided respondents with additional information on the patient:

> "on further questioning when the boy had recovered, he reported that the Chinese meal contained egg fried rice, beef, bean shoots, sesame, cashew nuts and peanuts. He has previously sometimes complained of itchy throat after eating chocolate and he had eczema in infancy worsened when eating egg. However, he has recently had egg without any reaction".

The following questions focused on patient management and additional investigations required, which were presented in the form of a multiple choice. Additional information on participants included age, gender, years of clinical experience, specialty, country and city of residence, clinical practice settings. Full version of the survey is available in the online supplementary document 1.

## Statistical analysis

Data were imported from the survey website into Microsoft Excel 2013, cleaned to detect any missing or invalid variables and then converted into the SPSS database. Statistical analysis was performed using IBM SPSS Statistics for Windows, V.22.0 (IBM, Armonk, New York, USA).

Statistical significance between the groups have been assessed using $x^2$ for categorical variables and Mann-Whitney U test for continuous variables. In order to evaluate any association between the demographic factors and ability to make the right diagnosis and treatment, logistic regression was used. Results were considered significant when p values were reported at a level <0.05.

## RESULTS
### Characteristics of the study population

The study was conducted using an online survey questionnaire tool. Of the 707 physicians accessed the survey, 315 (45%) responded to the clinical scenario and answered all the questions. Sixteen respondents reported that they had received training in allergy-immunology and have been excluded from the analysis, as we were not aiming to survey specialists, leaving the final sample size of 299.

The study population were predominantly women (85%), in their 50s (mean age of 47.7 years), with two-thirds (68%) of respondents worked in the paediatric field. More than half (57%) of the surveyed physicians work in primary care settings. Seventy per cent of the participants had practiced for more than 15 years. The demographic characteristics of study participants and physicians decided not to answer the survey are summarised in table 1.

## Respondents knowledge of anaphylaxis diagnosis and management

Thirty-three per cent of surveyed physicians diagnosed the presented patient with anaphylaxis. The most popular choice for first-line treatment options was intramuscular prednisolone (37%) followed by chloropyramine (26%), with only 15% of respondents suggested adrenalin (1:1000) intramuscular as a treatment of choice. Of those making the correct diagnosis, only 29% selected adrenalin as a first-line treatment (figure 1). Forty-six per cent of surveyed physicians considered salbutamol use as appropriate and 47% selected supplemental oxygen as a part of the necessary treatment. Data on respondents knowledge are presented in table 2.

Forty-eight per cent and 67% selected chest X-ray and spirometry, respectively, as required investigations. Most of the respondents considered sIgE (71%) and sIgG (63%) to food allergens testing as necessary tests, but only 23% suggested serum tryptase measurement.

Respondents working in secondary/tertiary clinics diagnosed patient with anaphylaxis significantly more often (p=0.04) when compared with primary care/private practice physicians. This was also the most important factor in the regression model, on adjustment for participants sex, age, specialty and clinical experience. Neither clinical experience nor specialty significantly influenced participants decision to use adrenalin. The medication choice was not associated with the participant characteristics (table 3).

Multivariate analysis outcomes (table 4) confirmed univariate analysis results showing that clinical setting is the most important factor influencing physicians ability

| Table 1 | Characteristics of the survey respondents and non-respondents. *: mean (SD) | | |
|---|---|---|---|
| **Characteristic** | **N (%) respondents** | **N (%) Non-respondents** | **p Value ($X^2$)** |
| Gender | | | |
| Male | 44 (15%) | 65 (17%) | |
| Female | 255 (85%) | 325 (83%) | 0.48 |
| Age | 47.7 (10.7)* | 49.0 (9.44)* | 0.06 |
| Specialty | | | |
| Paediatrics | 203 (68%) | 205 (53%) | |
| Other | 96 (32%) | 185 (47%) | **<0.01** |
| Clinical settings | | | |
| Primary care | 169 (57%) | 224 (57%) | |
| Secondary care | 67 (22%) | 94 (24%) | |
| Tertiary care | 36 (12%) | 42 (11%) | |
| Private practice only | 27 (9%) | 32 (8%) | 0.92 |
| Clinical experience (years) | | | |
| 1–5 | 29 (10%) | 25 (6%) | |
| 6–10 | 31 (10%) | 33 (9%) | |
| 11–15 | 30 (10%) | 54 (14%) | |
| >15 | 209 (70%) | 278 (71%) | 0.17 |

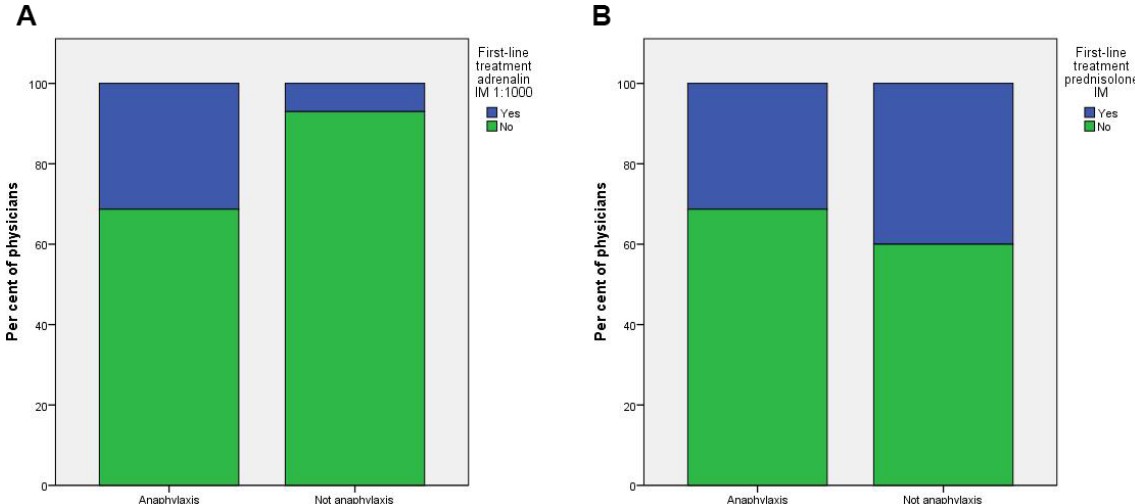

**Figure 1** Physicians' choice of adrenalin as a first-line treatment of anaphylaxis. This graph shows proportion of physicians to choose: A) intramuscular epinephrine 1:1000 or B) intramuscular prednisolone as a first-line treatment, depending on their choice of diagnosis.

to diagnose anaphylaxis. Those working in primary care/private practice make the correct diagnosis less often than in secondary/tertiary care hospitals OR 0.51 (95% CI 0.30 to 0.86). Participants diagnosing anaphylaxis were much more likely to treat patient with adrenalin OR 5.65 (95% CI 2.81 to 11.36). Prednisolone use was independent of participant characteristics. Two factors influencing chloropyramine prescription were age (younger age decreased the likelihood of prescribing) OR 0.97 (95% CI 0.93 to 0.99) and ability to diagnose anaphylaxis (right diagnosis decreased the likelihood of prescribing) OR 0.35 (95% CI 0.19 to 0.67), respectively.

## DISCUSSION

Anaphylaxis awareness has received more attention over the last decade. The findings of our study provide the first insight into knowledge of anaphylaxis diagnosis and management in Russia. Although recent reports from several countries[16] suggest that lack of knowledge in anaphylaxis diagnosis and management is not 'as great as previously published', differing patterns are seen in Russia. Sixty-seven per cent of respondents did not recognise anaphylaxis in a very clearly presented clinical scenario, for which anaphylaxis should be strongly considered. Approximately 1 in 10 professionals made an appropriate choice of adrenalin as a first-line treatment and only a third of those who rightfully diagnosed anaphylaxis provided appropriate medication. Given a response rate of 45%, our results are more likely to represent a best rather than worst-case scenario. It suggests that food allergic patients could be at increased risk of fatal reaction due to anaphylaxis in Russia and this danger is even higher considering lack of availability of AAIs in Russia.

Overall, only 10% of surveyed physicians were able to both diagnose and provide appropriate treatment for a child with food-induced anaphylaxis, when assessed using a clinical scenario survey instrument. These numbers

are much lower than in any of the previous studies up to date,[7 16 17] but corresponds well with real-life data, showing that anaphylaxis management is still far from satisfactory based on established recommendations. Analysis of a large anaphylaxis registry showed discrepancies between existing management guidelines and their implementation in clinical practice.[18] Similar outcomes have been recently confirmed from the European Anaphylaxis Registry database.[19]

Two studies from the USA included anaphylaxis scenarios with no skin rashes present,[7 13] as it is known that anaphylaxis can often manifest with no obvious cutaneous reaction. In our survey, we did not overcomplicate the clinical scenario, providing respondents with a combination of the cutaneous, cardiorespiratory, gastrointestinal and generalised symptoms. While the majority of the physicians recognised an allergic reaction in the clinical scenario, only 33% of respondents diagnosed the patient as having anaphylaxis. This outcome shows that most of the doctors are not aware of the anaphylaxis diagnostic criteria. Our clinical scenario clearly meets the anaphylaxis diagnostic criteria stated within the EAACI[4] and RAACI[14 15] guidelines. Poor EAACI guidelines awareness in Russia may be a result of a generally low level of English comprehension, but unawareness of RAACI guidelines is likely related to poor distribution and promotion.

Anaphylaxis recognition is the most important prerequisite for successful management, as physicians are more likely to administer adrenalin if they have correctly diagnosed the patient, as reported previously.[7] Our data are in agreement with the outcomes of previous studies. Among the factors potentially impacting participants' knowledge, we did not find any significant influence of primary specialty and/or clinical experience on physicians' ability to diagnose anaphylaxis. Respondents working in the secondary or tertiary centres possess better knowledge on anaphylaxis diagnosis, which may be explained by their

**Table 2** Respondents answers on anaphylaxis clinical scenario

| Outcomes | Responses | n=299 N (%) |
|---|---|---|
| *Diagnosis* What is the most likely diagnosis? | Anaphylaxis Not anaphylaxis | 99 (33) 200 (67) |
| *Treatment* Use adrenalin 1:1000 intramuscular as a first-line treatment | Yes No | 45 (15) 254 (85) |
| Use adrenalin 1:10 000 intramuscular as a first-line treatment | Yes No | 31 (10) 268 (90) |
| Use chloropyramine as a first-line treatment | Yes No | 79 (26) 220 (74) |
| Use prednisolone intramuscular as a first-line treatment | Yes No | 111 (37) 188 (63) |
| Uses prednisolone intramuscular in addition to other therapy | Yes No | 109 (37) 190 (63) |
| Oxygen supply | Yes No | 141 (47) 158 (53) |
| Salbutamol inhaler | Yes No | 139 (46) 160 (54) |
| Antihistamine (cetirizine) | Yes No | 11 (4) 288 (96) |
| Inhaled steroids | Yes No | 110 (37) 189 (63) |
| *Further investigations needed* Serum tryptase | Yes No | 69 (23) 230 (77) |
| Spirometry | Yes No | 199 (67) 100 (33) |
| Chest X-ray | Yes No | 144 (48) 155 (52) |
| Skin Prick Test (SPT) or specific immunoglobulin E (sIgE) testing to food allergens | Yes No | 213 (71) 86 (29) |
| Specific immunoglobulin G (sIgG) testing to food allergens | Yes No | 187 (63) 112 (37) |
| Stool ova and parasite exam | Yes No | 96 (32) 203 (78) |

daily exposure to highly specialised medical environment and involvement in academic activities.

Conversely chloropyramine use as a first-line treatment was inversely associated with anaphylaxis diagnosis. To what extent this is due to resort to medications which are freely available rather than a lack of knowledge cannot be determined but the latter is more likely given the poor responses to further investigation questions.

Intramuscular adrenalin is the drug of choice in the scenario of an anaphylactic event and should be injected as soon as possible.[18 19] It is evident that in our survey, most of the respondents preferred intramuscular prednisolone or oral chloropyramine use, irrespective of their

**Table 3** Multivariate predictive model for factors influencing physicians decision on anaphylaxis diagnosis and first-line management choice. Statistically significant difference (p<0.05) appear in bold

| Respondents characteristic | Diagnosed patient with anaphylaxis, % | OR (95% CI) | p Value | Used adrenalin 1:1000 intramuscular, % | OR (95% CI) | p Value | Used prednisolone intramuscular, % | OR (95% CI) | p Value | Used chloropyramine, % | OR (95% CI) | p Value |
|---|---|---|---|---|---|---|---|---|---|---|---|---|
| *Clinical settings* Primary care/private practice | 29 | 0.59 (0.36 to 0.98) | **0.04** | 13 | 0.68 (0.35 to 1.29) | 0.23 | 41 | 1.60 (0.96 to 2.66) | 0.07 | 27 | 1.18 (0.69 to 2.05) | 0.54 |
| Secondary and/or tertiary care | 41 | | | 18 | | | 30 | | | 24 | | |
| *Clinical experience (years)* 1–10 | 32 | 0.92 (0.50 to 1.69) | 0.79 | 15 | 0.99 (0.45 to 2.20) | 0.99 | 38 | 1.07 (0.60 to 1.91) | 0.83 | 28 | 1.13 (0.60 to 2.12) | 0.71 |
| >10 | 34 | | | 15 | | | 37 | | | 26 | | |
| *Specialty* Paediatrics | 34 | 1.22 (0.72 to 2.05) | 0.46 | 16 | 1.36 (0.67 to 2.77) | 0.40 | 39 | 1.36 (0.82 to 2.28) | 0.23 | 29 | 1.43 (0.81 to 2.53) | 0.22 |
| *Other specialities* | 30 | | | 13 | | | 32 | | | 22 | | |

**Table 4** Diagnosis and first-line treatment among different respondent groups. X$^2$ test is used for the group comparison. Statistically significant difference (p<0.05) appear in bold

| | Anaphylaxis diagnosis OR 95% CI | p Value | Intramuscular adrenalin OR 95% CI | p Value | Prednisolone OR 95% CI | p Value | Chloropyramine OR 95% CI | p Value |
|---|---|---|---|---|---|---|---|---|
| Sex | 0.75 (0.36 to 1.57) | 0.45 | 0.71 (0.24 to 2.10) | 0.54 | 1.69 (0.85 to 3.83) | 0.14 | 1.49 (0.70 to 3.15) | 0.30 |
| Age | 0.97 (0.95 to 1.00) | 0.10 | 0.98 (0.94 to 1.03) | 0.42 | 1.00 (0.98 to 1.04) | 0.61 | 0.97 (0.93 to 0.99) | **0.04** |
| Specialty | 1.34 (0.77 to 2.35) | 0.31 | 1.30 (0.60 to 2.83) | 0.51 | 1.43 (0.82 to 2.50) | 0.20 | 1.59 (0.85 to 2.97) | 0.15 |
| Clinical settings | 0.51 (0.30 to 0.86) | **0.01** | 0.74 (0.36 to 1.50) | 0.40 | 1.55 (0.90 to 2.68) | 0.11 | 0.95 (0.52 to 1.73) | 0.86 |
| Clinical experience | 1.33 (0.59 to 2.99) | 0.49 | 1.39 (0.45 to 4.31) | 0.57 | 0.94 (0.43 to 2.05) | 0.88 | 1.93 (0.81 to 4.61) | 0.14 |
| Anaphylaxis diagnosis | NA | NA | 5.65 (2.81 to 11.36) | **<0.01** | 0.73 (0.44 to 1.24) | 0.24 | 0.35 (0.19 to 0.67) | **<0.01** |

specialty, clinical experience and/or clinical settings. Prednisolone or other corticosteroids can be used for biphasic anaphylaxis prevention, which accounts for around 20% of anaphylactic cases,[20] and may reduce the risk of late-phase respiratory symptoms,[4] but it does not resolve acute anaphylaxis. First-generation anti-histamines are not an optimal choice for anaphylaxis management, especially when patient is lethargic, as they may cause drowsiness and mask respiratory symptoms.[14]

Beta-2 agonists can be used during anaphylaxis to assist in the event of bronchoconstriction and the patients should be provided with supplemental oxygen,[4] but only half of respondents selected these options as a part of required management.

The majority of respondents choose SPT/sIgE and/or sIgG testing to food allergens, to detect a possible trigger of reaction. The number of physicians prioritising SPT/sIgE over sIgG testing to food was little different. These results suggest lack of differentiation between sIgE and sIgG among Russian physicians. Recently, a number of position papers has been released, warning practitioners against IgG testing use in the diagnosis of food allergy,[21–23] but these papers are not easily available in Russia and no relevant statements has been provided by RAACI. Approximately one in four respondents considered serum tryptase as a useful test in the case scenario. Routine evaluation of serum tryptase levels is still a grey area, but recent data suggest that it is particularly useful in severe anaphylactic events due to drugs and wasp/bee venom reactions. However, tryptase is not raised in many cases of food-induced anaphylaxis other than if milk is a causative allergen.[24]

There were several limitations to the study. The survey is not standardised and validated. It was distributed via VrachiRF website, which can be considered as one of the most popular sources for Russian physicians. Low response rate is a recognised issue in survey research, which potentially may lead to selection bias. Of the physicians accessed the survey 45% completed it, which is not ideal but slightly above average for online surveys.[25] It is not known how many doctors out of half a million subscribers accessed any part of the website during the month that the questionnaire was active. Respondents tended to work in paediatric field more often than non-respondents, probably due to the case scenario describing anaphylaxis in a child.

Clinicians using medical websites and responding to a competence questionnaire may possess better knowledge on a wide range of medical topics, meaning that survey outcomes do not necessarily provide an image of an average doctor practicing in this country. Lack of correct answers does not always automatically translate to wrong clinical decisions in a real-life situation. There are 162 000 licensed adult physicians and 65 000 practicing paediatricians in Russia, which does not allow data extrapolation to the whole Russian medical fraternity. As the survey was presented in Russian, results cannot be generalised to medical professionals

living in other countries. However, we believe that our results are closer to a best-case interpretation and therefore are even more concerning.

Although limitations are clear, results of this survey suggest a common deficit of knowledge on anaphylaxis diagnosis and management in Russia. Despite anaphylaxis guidelines being easily accessible via EAACI and RAACI websites, they are not promoted well enough among Russian health professionals. Our data highlight the need for educational programme development and changes to the undergraduate and postgraduate medical education curriculum. This should lead to a better care quality for children with food allergy at risk of anaphylactic events and may potentially reduce morbidity and mortality. Furthermore, detection of existing gaps in anaphylaxis diagnosis and management is important[16] at a time when food allergy prevalence is rising, particularly in the paediatric population.

**Acknowledgements** We would like to thank physicians who kindly agreed to complete the survey and VrachiRF.ru website team for their incredible help with the survey.

**Contributors** DM, MT, IK and JW designed a questionnaire. AA and AP provided an independent evaluation of the survey instrument and assisted with the presurvey pilot testing. DM and MT conceived the study, designed the methods and coordinated data collection. DM wrote the first draft of the paper. MT and JW contributed to the writing of the paper.

**Funding** This research received no specific grant from any funding agency in the public, commercial or not-for-profit sectors. However, JW is funded through the National Institute of Health Research (NIHR), Collaboration for Leadership in Applied Health Research and Care for NW London. The views expressed in the paper are those of the authors and are not necessarily those of the National Health Service, the NIHR or the Department of Health.

**Competing interests** None declared.

**Ethics approval** R&D office, Speransky Hospital, I.M. Sechenov First Moscow State Medical University, Moscow, Russia.

**Provenance and peer review** Not commissioned; externally peer reviewed.

**Data sharing statement** The Excel data set is available from corresponding author on request.

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
