## [Reviewer comments · BMJ Open]

ARTICLE DETAILS

TITLE (PROVISIONAL)	A national survey of Russian physicians' knowledge of diagnosis and management of food induced anaphylaxis
AUTHORS	Munblit, Daniel; Treneva, Marina; Korsunskiy, Ilya; Asmanov, Alan; Pampura, Alexander; Warner, John O.

VERSION 1 - REVIEW

REVIEWER	Jose Colleti Junior Hospital Santa Catarina, São Paulo, Brazil
REVIEW RETURNED	23-Jan-2017

GENERAL COMMENTS	The study is up-to-date and addresses an important subject. The English spelling and grammar should be reviewed in general.
---

REVIEWER	Prof. Dr. med. Margitta Worm Department of Dermatology and Allergy, Charité-Universitätsmedizin Berlin, Germany
REVIEW RETURNED	03-Mar-2017

GENERAL COMMENTS	The presented paper by Munblit et al. presents data obtained from a questionnaire which was answered by physicians in Russia. The purpose of a case based survey was to evaluate the physicians knowledge on anaphylaxis diagnosis and treatment. The data indicates that the case was largely underdiagnosed and the suggested treatment was not according to the guidelines. Such data in particular regarding the underrepresented use of adrenaline has been published previously (Grabenhenrich, PLoS ONE 2012 and Grabenhenrich, JACI 2016). This data should be included in the discussion and the citations should be added. I have some general concerns about the data. First, the rate of answering doctors is not easy to rate as the number of licensed doctors in Russia is not mentioned. I suggest to include this and also how many doctors in Russia are pediatricians. The differentiation between the primary and secondary and/or tertiary care is interesting, however based on the finally very low numbers of doctors the conclusions drawn by the data should be taken with great care. Interesting is the observation that a large proportion of the doctors answered that SPT or IgE testing to food allergens should be performed, therefore I see a controversial response as it seems that food allergy as a diagnosis or DD has been recognized. Language aspects may have influenced the results as well. I cannot see whether the questionnaire was in Russian or in English and if it was in Russian whether it has been back translated before use. This
--

	are small methodological aspects but might be of importance.
--	--

REVIEWER	Simon Craig Monash Health and Monash University, Australia
REVIEW RETURNED	12-Mar-2017

GENERAL COMMENTS	Thank you for the opportunity to review this paper reporting a survey of Russian doctors regarding the diagnosis and management of food-related anaphylaxis in children. Introduction Sets the scene for the survey, and provides a reasonable rationale for conducting the study. There are some minor typographic errors, which will be able to be corrected at the proofing stage. Methods Some further information about the VrachiRF website (or the number of people that regularly access it) would be helpful to provide some more information about the study population. Is the website available to all types of doctors working in Russia? Does the site send regular emails, or does it rely on visits only? If it sends out emails, how often, and to how many subscribers? If only posted on the website, how long was it posted for? 1 week? 4 weeks? It would also be helpful to be able to view the whole survey (perhaps as a supplementary file) Results It would be interesting to see the major categories of diagnosis for those respondents who did not diagnose anaphylaxis. Table 3 should present odds ratios (with 95% confidence intervals) as well as P values. Discussion and conclusion Reasonable points, although further information regarding possible response rate would be helpful.
---

VERSION 1 – AUTHOR RESPONSE

Reviewer: 1

Reviewer Name: Jose Colleti Junior

Institution and Country: Hospital Santa Catarina, São Paulo, Brazil

Competing Interests: None declared

The study is up-to-date and addresses an important subject. The English spelling and grammar should be reviewed in general.

Response: Thank you for your kind words and high opinion of our survey.

Reviewer: 2

Reviewer Name: Prof. Dr. med. Margitta Worm

Institution and Country: Department of Dermatology and Allergy, Charité-Universitätsmedizin Berlin, Germany

Competing Interests: none

The presented paper by Munblit et al. presents data obtained from a questionnaire which was answered by physicians in Russia. The purpose of a case based survey was to evaluate the physicians knowledge on anaphylaxis diagnosis and treatment. The data indicates that the case was largely underdiagnosed and the suggested treatment was not according to the guidelines.

Such data in particular regarding the underrepresented use of adrenaline has been published previously (Grabenhenrich, PLoS ONE 2012 and Grabenhenrich, JACI 2016). This data should be included in the discussion and the citations should be added.

Response: Thank you for this comment. Additional paragraph on this matter has been added in the discussion section.

I have some general concerns about the data.

First, the rate of answering doctors is not easy to rate as the number of licensed doctors in Russia is not mentioned. I suggest to include this and also how many doctors in Russia are pediatricians. The differentiation between the primary and secondary and/or tertiary care is interesting, however based on the finally very low numbers of doctors the conclusions drawn by the data should be taken with great care. Interesting is the observation that a large proportion of the doctors answered that SPT or IgE testing to food allergens should be performed, therefore I see a controversial response as it seems that food allergy as a diagnosis or DD has been recognized. Language aspects may have influenced the results as well. I cannot see whether the questionnaire was in Russian or in English and if it was in Russian whether it has been back translated before use. This are small methodological aspects but might be of importance.

Response: Many thanks for your comments. We added data on the number of licensed doctors and paediatricians in Russia. We are in full support of your opinion with regards to selection bias due to small numbers and included this into discussion section.

The observation that a large proportion of the doctors answered that SPT or IgE testing to food allergens should be performed can be explained by additional information with regards to the patient, which we provided to the respondents (we added this into the methods section, to make it clear to the reader). We assume that these details led respondents to select allergy testing.

We added full survey questionnaire as a supplementary material as suggested.

Language was not an issue for the survey respondents as physicians were provided with an instrument in Russian. We provide information within the methods section "we developed a case based survey in Russian" and in the limitation part of discussion section "As survey was presented in Russian" to inform the reader that this survey was undertaken in the respondents mother-tongue.

Reviewer: 3

Reviewer Name: Simon Craig

Institution and Country: Monash Health and Monash University, Australia

Competing Interests: None declared

Thank you for the opportunity to review this paper reporting a survey of Russian doctors regarding the diagnosis and management of food-related anaphylaxis in children.

Introduction

Sets the scene for the survey, and provides a reasonable rationale for conducting the study. There are some minor typographic errors, which will be able to be corrected at the proofing stage.

Response: Thank you for your kind words. We proofread the manuscript and corrected a few

typographic errors.

Methods

Some further information about the VrachIRF website (or the number of people that regularly access it) would be helpful to provide some more information about the study population. Is the website available to all types of doctors working in Russia? Does the site send regular emails, or does it rely on visits only? If it sends out emails, how often, and to how many subscribers? If only posted on the website, how long was it posted for? 1 week? 4 weeks?

Response: Thank you for this suggestion. We provide some additional information with regards to the website and the survey.

It would also be helpful to be able to view the whole survey (perhaps as a supplementary file)

Response: Thank you for the comment. We added full survey questionnaire as a supplementary file, as suggested.

Results

It would be interesting to see the major categories of diagnosis for those respondents who did not diagnose anaphylaxis.

Table 3 should present odds ratios (with 95% confidence intervals) as well as P values.

Response: Thank you for this comment. Odds ratios (with 95% confidence intervals) were added in Table 3.

Discussion and conclusion

Reasonable points, although further information regarding possible response rate would be helpful.

Response: Thank you for this comment. Critical appraisal of the response rate has been added into the limitations part of the discussion section. We also provide information on non-responders in Table 1.

VERSION 2 – REVIEW

REVIEWER	Simon Craig Monash Medical Centre Emergency Department. Melbourne, Australia
REVIEW RETURNED	30-Mar-2017

GENERAL COMMENTS	Thank you for the opportunity to review a revised manuscript. My only concern is the stated response rate is inaccurate. The website is viewed (and sends emails to) approximately half a million subscribers. Therefore, approximately half a million subscribers were sent details of the survey. Around 700 people accessed the survey, and 315 responded to the questions. Therefore, there were 315 responses from approximately half a million subscribers.
--

VERSION 2 – AUTHOR RESPONSE

Reviewer 3

Many thanks for your comment.

The website sends newsletters to the subscribers, but they are news from the medical world. Our survey has not been promoted or adverted, apart from being placed to the website for one month. Unfortunately, it is impossible to get information with regards to the number of people accessed different sections of the website, we only know how many people accessed the survey itself. We consistently state that "From a total of 707 of physicians accessed in the survey, 315 (45%) responded to the clinical scenario." throughout the manuscript (abstract, results and discussion sections).

We made the following changes to the manuscript, following your suggestions:

Within the methods section we made a minor change, saying that "VrachiRF placed a survey on the website for 4 weeks (between February and March 2016)" to avoid misconception. Otherwise readers may assume that it was emailed to the subscribers, which was not the case.

We also made a change to the limitations section, providing a more detailed and precise explanation, saying that "Of the physicians accessed the survey 45% completed it, which is not ideal but slightly above average for online surveys. It is not known how many doctors out of half a million subscribers accessed any part of the website during the month that the questionnaire was active."

This however, as we have indicated in the discussion most likely presents a best rather than worst case scenario. Thus in the final analysis the proportion answering is immaterial.

VERSION 3 – REVIEW

REVIEWER	Simon Craig Monash Medical Centre, Australia
REVIEW RETURNED	22-May-2017

GENERAL COMMENTS	Thank you for addressing the earlier queries. There are a few minor suggestions which should be addressed prior to final acceptance: Line 50 "accessed in the survey" should be changed to "accessing the survey" Main text – line 182 - same as above: "accessed the survey" should be changed to "accessing the survey" Main text – line 196 - 33% of respondents (not surveyed physicians) Line 322 – remove the word "be"
---